# Democratizing social media for health research: LLM-powered data analytics platform for NCDs

Ratchanont Thippimanporn◉, Wuttichai Khamna◉,
Kannika Wiratchawa, Thanapong Intharah◉*

Visual Intelligence Laboratory, Department of Statistics, Faculty of Science, Khon Kaen University, Thailand

◉ The authors contributed equally (Co-first Authors).

* thanin@kku.ac.th

**Data availability statement:** The data used in our research is publicly accessible at https://doi.org/10.7910/DVN/HRSM09. The

## Abstract

Despite over 41 million annual deaths from Non-Communicable Diseases (NCDs) globally, predominantly in low and middle-income countries, public access to relevant information from social media is hindered by restrictive licensing of existing social listening tools. This study introduces NCDs Listener, an open-source tool designed to simplify the extraction, summarization, and visualization of NCD-related knowledge from social media comments (Facebook and Reddit posts) in both English and Thai. The tool utilizes keyword matching and the BERT model for knowledge extraction, followed by descriptive statistical analysis. A generative AI model, specifically Google Gemini 2.0 Flash as per the saved information, summarizes this extracted knowledge into human-readable sentences, focusing on medical and healthcare insights. Preliminary results indicate that NCDs Listener improves dashboard comprehension for both general users and data scientists, with the general users showing higher comprehension. Furthermore, both user groups preferred medically focused generative AI summaries over general summaries (p-value <0.001). These findings suggest that NCDs Listener not only provides immediate insights but also establishes a foundation for advanced data analysis, fostering new opportunities for understanding complex social phenomena and predicting emerging trends. The source codes are available at the project page: https://ratchanontt.github.io/NCDsListenerWebpage/.

## Introduction

Non-Communicable Diseases (NCDs) are chronic conditions that cannot be spread from person to person and are not caused by infectious agents like bacteria or viruses. Instead, NCDs often result from unhealthy lifestyles and typically develop over long periods. Globally, NCDs are responsible for 41 million deaths annually, accounting for 74% of all deaths. Of these, 17 million individuals die before the age of 70, with 86% of these premature deaths occurring in low- and middle-income countries. 77% of all NCD deaths are concentrated in these regions. Cardiovascular diseases claim the highest toll, with 17.9 million deaths each year, followed

repository contains: (1) anonymized raw data used for model creation, (2) query data for graph and variable analysis, and (3) query data supporting the communication functions of both the Dashboard and the Generative AI. All datasets have been thoroughly anonymized before publication.

**Funding:** The author(s) received no specific funding for this work.

**Competing interests:** The authors have declared that no competing interests exist.

by cancers (9.3 million), chronic respiratory diseases (4.1 million), and diabetes (2.0 million, including kidney disease associated with diabetes). Collectively, these four disease groups account for over 80% of all premature NCD-related deaths [1].

Decisions regarding the treatment or care of Non-Communicable Diseases (NCDs) often raise numerous concerns, such as identifying the behaviors that contribute to these diseases, recognizing symptoms before and after diagnosis, and determining the best course of treatment. Despite the abundance of available information on NCDs, relatively few platforms present this knowledge in a friendly, accessible way for patients and caregivers. Those with firsthand experience either living with NCDs or caring for someone who does are often best suited to address these concerns. In recent years, social media has become essential for connecting these individuals, transforming how patients learn about their conditions, forming peer support networks, and sharing personal experiences [2].

Social listening studies have primarily focused on developing systems that analyze sentiment and track trending topics [3]. A considerable amount of research has been conducted using social listening tools. However, the tools, such as Brand24 and YouScan [4], require expensive monthly subscriptions, which makes them inaccessible to the general public or independent researchers. Furthermore, these tools are often designed to cater to large organizations, limiting their applicability to individuals who wish to analyze social media data on a personal level [4]. We developed an open-source tool to analyze public social media posts about NCDs and present the extracted insight as a dashboard. This helps the general public and researchers study how the NCD community shares their real-life experiences while reducing the time needed to find relevant information and improving public access to understanding these conditions.

This study focuses on the development of the NCDs Listener web application, a social media listening tool designed to extract and analyze public sentiment from posts related to NCDs. The extracted data is presented through a customizable dashboard. The research employs Natural Language Processing (NLP) and Machine Learning techniques to derive meaningful insights from social media data. Additionally, Large Language Models (LLMs) are integrated to enhance the social media listening process. Furthermore, this study examines user needs and assesses the comprehension of user groups regarding the dashboard results and generalizations generated by Large Language Models. The target users of this tool can be categorized into two primary groups: the general public, individuals without Data Science Expertise, and Researchers or Data Scientists.

The researcher will randomly select two sets of users from the target groups, as defined in the preceding paragraph. The first set will review and select the desired variables and chart types before the dashboard is created. The second set will then evaluate the dashboard's communication ability by reviewing the results generated based on the selections made by the first set. Additionally, the second set will assess the communication ability of generative AI by reviewing multiple example summaries produced using different approaches specified by the researcher. The obtained results will serve as the basis for selecting an approach to summarize the overall characteristics of opinions based on user requirements.

## Related work

In this section, we delve into existing research on social listening for non-communicable diseases (NCDs) and explore the diverse applications of social media listening within this context. Our discussion will draw upon relevant studies that have investigated these two key areas.

## Social listening for non-communicable diseases

Many researchers are working on expanding knowledge on social media listening for Non-Communicable Diseases in the literature, and some key contributions are providing support for identifying user behaviors and situations in various cases worldwide. Some of the essential papers are included in this section. For disease-specific social listening studies, various social media listening tools and data collection and analysis techniques,

A. Rodrigues *et al.* [5] focused on European social media conversations to understand the experiences of lung cancer patients, caregivers, and healthcare professionals. Using Talkwalker and SocialStudio.

J. Chauhan *et al.* [6] analyzed the experiences of melanoma patients across 14 European countries using SocialStudio and Talkwalker. They identified significant impacts on patients' daily lives and emotions.

M. Mazza *et al.* [7] used social listening on Twitter, patient forums, and blogs to explore metastatic breast cancer patient experiences.

Z. Perić *et al.* [8] investigated GVHD patient needs and lifestyles across Europe using Talkwalker to collect data from Twitter, Facebook, Instagram, and YouTube. The research employed a combination of quantitative and qualitative methods to analyze the quality of life, treatment efficacy, and unmet needs.

Most insights from these studies focus on variables and result presentation formats, which help shape our tool development, like deciding what data to store and how to display results in graphs or dashboards. However, these studies relied on data from existing tools, such as Talkwalker and SocialStudio, with manual analysis performed by researchers. This sets our work apart: We offer a clear approach to automating data collection and computational analysis. Our research also incorporates the latest large language model techniques to summarize extracted knowledge into concise, human-readable paragraphs.

## Applications of social media listening

Different studies have employed various tools and techniques to analyze social media data, with a focus on understanding public sentiment and opinions. For analysis techniques employing NLP,

K. H. Manguri *et al.* [9] analyzed Twitter data from the COVID-19 pandemic using the hashtags #coronavirus and #COVID-19. They applied NLP and Sentiment Analysis.

J. Burzynska, A. Bartosiewicz, and M. Reka [10] used the SentiOne tool for NLP to analyze social media data on COVID-19 in Poland. Their analysis revealed an increase in public discussions and information sharing as the pandemic progressed. For analysis techniques utilizing Latent Dirichlet Allocation (LDA) and Topic Modeling,

C. C. Shoults *et al.* [11] employed LDA and t-SNE to analyze social media discussions about telemedicine on Reddit and Twitter,

A. C. Sanders *et al.* [12] conducted sentiment analysis and clustering of Twitter data to study public opinions on face masks during COVID-19. Additionally, quantitative and qualitative analysis, along with bot detection, were employed for social media data analysis;

G. Spitale, N. Biller-Andorno, and F. Germani [13] analyzed Telegram conversations about the Green Pass in Italy using both quantitative and qualitative methods.

Most research on social media listening leverages technologies such as Python for data aggregation, often utilizing packages like Tweepy, which typically necessitate API access or reliance on social media platform functionalities. These studies commonly employ Natural Language Processing (NLP) techniques, including tokenization, stopword removal, sentiment analysis, and topic modeling. In contrast, our open-source NCD Listener tool utilizes Python

packages such as Selenium and Beautiful Soup 4, providing out-of-the-box solutions that require only a URL for data acquisition. We also utilize advanced NLP techniques, encompassing tokenization, lemmatization, normalization, and BERT for topic classification. Nevertheless, the insights derived from this research are primarily conveyed through descriptive statistics and dashboard visualizations as well.

## Criteria for formulating analytical questions

Based on existing research, we found that there are various criteria for measuring the performance of dashboards and generative AI. However, we are particularly interested in the dashboard performance criteria proposed by S. Almasi *et al.* [14]. The researchers examined the criteria used to analyze dashboards across various studies by investigating the questions and response formats. They then summarized the criteria used in the analysis, including factors such as Usefulness, Operability, Learnability, and others.

We are also interested in the generative AI performance criteria proposed by J.-L. Peng *et al.* [15]. The researchers examined performance measures for generative AI from various studies to evaluate the performance of Large Language Models (LLMs). They discussed how to assess the capabilities of LLMs to determine the tasks for which they should be responsible. They proposed a two-step framework consisting of "core capabilities" and "representative capabilities" to explain how LLMs can be applied according to specific capabilities, along with the evaluation method for each step.

Therefore, we define the dashboard performance criteria in terms of content criteria, which are divided into two components: the Appropriateness of the amount of information for task performance and the Quality of the information provided for dashboard communication effectiveness. For the generative AI performance criteria, we define the core ability criteria, which consist of 3 parts: Reasoning, Societal Impact, and Domain Knowledge, for the selection of generative AI-based summarization approaches.

A previous work [16] focused on the development of a social media listening tool for Non-Communicable Diseases (NCDs). This tool integrated data collection from social media, knowledge extraction via keyword matching and machine learning for comment categorization, and presented results through a dashboard and Generative AI-powered summaries of NCD-related discussions. However, the absence of user group testing meant the developed tool did not directly align with user needs. Consequently, the objective of the current work is to rigorously test and further refine the system to optimize its alignment with user requirements.

Moreover, this research introduces a tool designed to minimize financial costs, time expenditure, and human resources required for the automated collection, analysis, and presentation of Non-Communicable Disease (NCD) information from social media. This addresses a significant gap, as much existing research in Social Listening for NCDs [5–8] typically necessitates considerable resources for data aggregation, analysis, and visualization. While most studies on Applications of Social Media Listening [9–13] have achieved automated data processing and presentation, they often still demand programming expertise for utilization, and some are not directly focused on health content [9,12,13]. This limitation frequently leads to challenges in health data analysis, often yielding only sentiment insights rather than in-depth information such as symptoms or treatment methodologies. Consequently, this tool significantly advances studies in information science, health data science, and related fields, broadening their potential for wider application.

## Methodology and development of the NCDs listener tool

This study extends previous work by introducing the NCDs Listener, a social listening tool. The code, model development data, and evaluation data can be accessed through the project repository [16,17]. We conducted the Mann-Whitney U test using Statistical Package for the Social Sciences (SPSS) version 2.6. The NCDs Listener was designed to analyze information about Non-Communicable Diseases from social networks. The research in [16] focused on social network data preparation and analysis, but it still lacked user studies to evaluate design choices and system performance. Thus, this study complements the existing work by focusing on the development of a system that better aligns the tool with user needs. This section outlines the steps involved in developing the NCDs Listener tool, beginning with system design, artificial intelligence models for insight extraction, and the design process for the dashboard. It further details evaluating dashboard communication effectiveness and selecting generative AI-based summarization approaches.

### Ethics statement

This study was conducted in strict adherence to ethical guidelines, with informed consent obtained from all participants in written forms before their involvement. The Human Ethics Committee of Khon Kaen University, Thailand, based on the ethics of human specimen experimentation of the National Research Council of Thailand, confirmed that formal ethics approval was not required as the research primarily constituted a quality assurance measure with anonymous survey collection (Application Number: HE682030). Confidentiality and data privacy were rigorously maintained throughout the research process, and all data were analyzed anonymously to protect participants' privacy and well-being.

### Systems design

The NCDs Listener system was developed using Python to collect, extract knowledge, and summarize social media posts related to NCDs. It helps turn these posts into meaningful insights. The system implements a 5-step workflow:

1. Data Scraping Step: When a user provides a URL for a social media post, the NCDs Listener tool uses web scraping to extract relevant data. The system accesses the URL and then scans the content based on a set of predefined HTML tags, as specified by the researcher for the specific platform. This process automatically gathers data (the user does not need to define any HTML tags), including information that may be hidden within the page structure. This data is then stored in a four-column table: name, comments, likes, and replies. The Selenium Python package [18] was used to reveal and access this data, and Beautiful Soup 4 [19] collected information through the specified tags.

2. Data Preprocessing Step: The system cleanses the collected data by removing duplicate comments, filtering out extremely short (less than or equal to 5 words) text entries, and anonymizing personal names by replacing them with Name(n), where "n" represents the comment number. Our privacy-preserving methods involved removing names from both the database and the web interface, preventing users from extracting and reusing this identifying information. The preprocessed data can be exported in CSV format for future analysis. The tool only scrapes data that is shared publicly on the internet.

3. Extracting Knowledge Step: The system employs Natural Language Processing (NLP) techniques, including tokenization, stop word removal, and normalization (or lemmatization), utilizing Python packages NLTK [20] and PythaiNLP [21]. Keyword matching

is applied to identify knowledge about Non-Communicable Diseases (NCDs) specifically. Additionally, the application of a machine learning model further aids in categorizing comments.

4. Data Adjustment Step: Users can add diseases and symptoms they wish to include or exclude those they do not want. The system then filters the comments based on the specified diseases and symptoms, tailoring the data to the user's requirements. Upon obtaining the user's desired information, the system proceeds to process all relevant text comments (aligned with user specifications). These comments are then converted into text vectors using the Embedding-001 model [22]. This vectorization creates a knowledge base, enabling the Generative AI to produce medically-focused summaries.

5. Data Visualization Step: The enhanced data is summarized and displayed on an interactive dashboard. Additionally, a comprehensive summary is generated by a Google Gemini 2.0 Flash model [23]. To ensure the accuracy of this generated summary, the Retrieval-Augmented Generation (RAG) [24] principle is employed, utilizing the previously vectorized data as its knowledge base. Users can also explore the data from a summary statistics table about the post or export the results to a PDF. The workflow is presented in Fig 1.

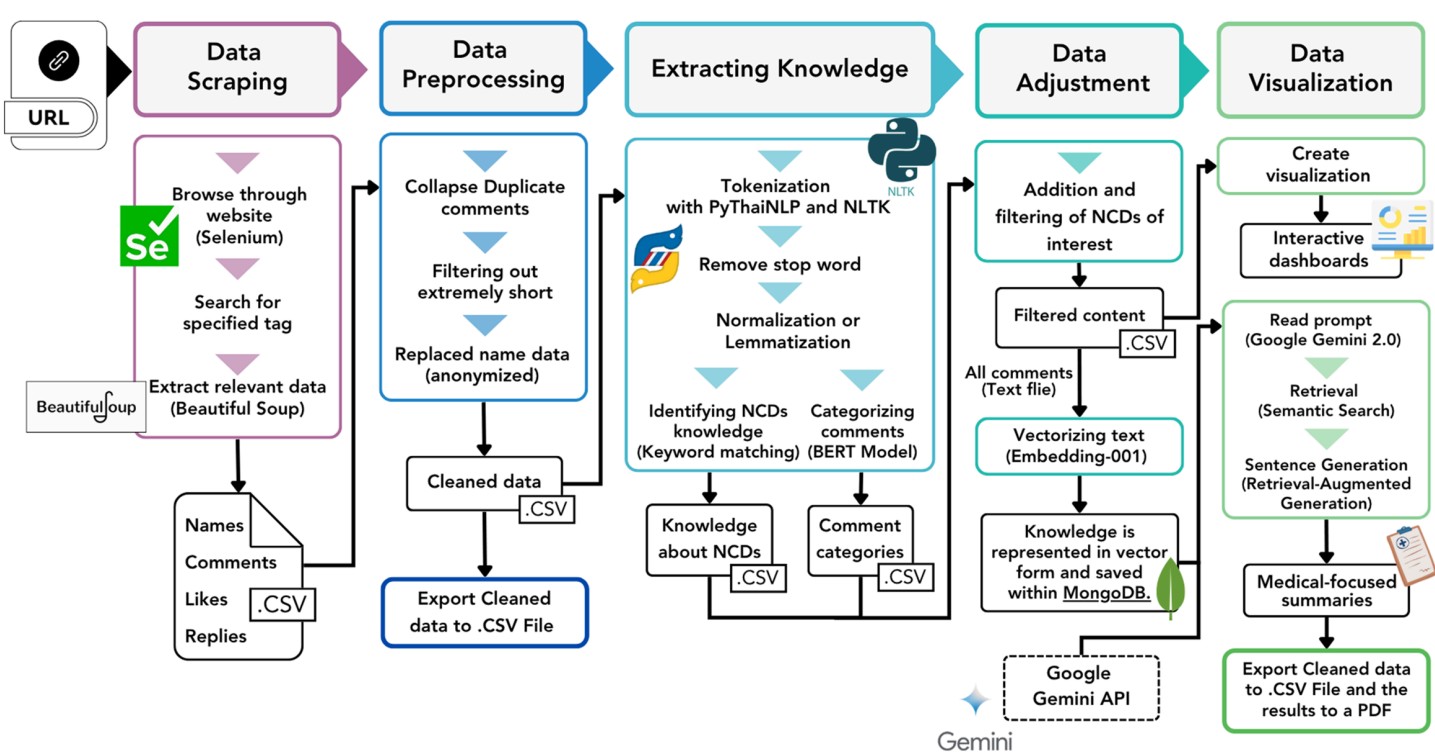

**Fig 1. System design and workflow for social media–based NCD monitoring and summarization.** The workflow consists of five stages: (1) Data Scraping – Social media data are collected using Selenium and BeautifulSoup, extracting relevant elements (names, comments, likes, replies) into .CSV files. (2) Data Preprocessing – Duplicate and extremely short comments are removed, and personal identifiers anonymized. (3) Extracting Knowledge – Cleaned text is tokenized, stop words removed, and normalized or lemmatized; NCD-related content is identified via keyword matching, and comment categories classified using a BERT model. (4) Data Adjustment – Content is refined for NCDs of interest, vectorized (Embedding-001), stored in MongoDB, and processed with the Google Gemini API. (5) Data Visualization – Interactive dashboards present results, with retrieval-augmented generation producing summaries and medical-focused narratives. Final datasets and reports are exportable in .CSV and PDF formats.

## Artificial intelligence models for insight extraction

In this work, we employed 2 AI models in the insight extraction. First, machine learning was used for topic classification. Second, LLMs were used for content summarization.

**Machine learning for topic classification.** This section details the methodology for the development and evaluation of the BERT model [24] for topic classification. It outlines the complete model development pipeline and presents a comprehensive performance analysis, including an interpretation of the confusion matrix (presented in Fig 2) to assess classification accuracy.

### BERT Model Output Categories

The fine-tuned BERT model serves as an automated classifier, designed to categorize social media comments with high efficacy. This classification enables a structured analysis of public discourse surrounding NCDs. The model effectively assigns each comment to one of the following three predefined categories:

*Non-informative*: Comments that are not related to NCDs or lack meaningful information, such as simply stating the name of a disease or unclear descriptions of symptoms. This category is used to filter out such data, thereby improving the accuracy and relevance of the final results.

*Sharing experience*: Comments that share personal experiences with diseases, including symptoms (before or after diagnosis), treatments, and how the disease has affected the quality of life.

*Inquiring*: Comments that inquire about various aspects of NCDs, such as symptoms, treatments, or disease impacts.

**The model development process was conducted in 4 sequential stages:**

1. **Step 1: Dataset Collection and Annotation** A dataset comprising over 3,000 comments related to Non-Communicable Diseases (NCDs) was extensively collected from

**Fig 2. Confusion matrices of classification performance for the Thai and English models.** The matrices illustrate the performance on the hold-out test sets for the Thai model (left), evaluated on 930 instances (30% of the Thai test data), and the English model (right), evaluated on 1,036 instances. The diagonal cells represent the number of correctly classified instances for each class, while off-diagonal cells indicate misclassifications. Both models demonstrate strong predictive accuracy, particularly for the "Sharing experience" and "Non-informative" classes. A comprehensive breakdown of performance metrics, including precision, recall, and F1-scores, is provided in Table 1 (Thai) and Table 2 (English).

social media platforms. This dataset was then manually categorized into three distinct classes: "Non-informative", "sharing experience", and "inquiring". To ensure high-quality and reliable ground-truth data, the final label for each comment was determined by achieving a consensus among three expert annotators.

2. **Step 2: Data Partitioning and Preprocessing** For robust model evaluation, the annotated dataset was partitioned into a 70% training set and a 30% held-out test set using the train_test_split function from the scikit-learn library in Python. This division ensures that the model's final performance is assessed on unseen data, providing an unbiased measure of its generalization capability. To mitigate the effects of class imbalance within the training data, an oversampling technique was applied using the "imbalanced-learn" Python package. Subsequently, the raw text underwent a data preparation pipeline involving two key steps: tokenization and stop-word removal. These procedures were performed using the PyThaiNLP library for Thai text and the Natural Language Toolkit (NLTK) for English text.

3. **Step 3: Model Selection via Transfer Learning** We employed a transfer learning approach to leverage the extensive linguistic knowledge encapsulated in large pre-trained language models. For the Thai language classifier, the "wangchanberta-base-att-spm-uncased" model was selected as the foundational architecture [26]. For the English language classifier, the "bert-large-uncased" model was utilized [27].

4. **Step 4: Model Fine-Tuning and Training** Separate models for the Thai and English languages were fine-tuned using identical hyperparameters. Each BERT-based model was trained with a maximum sequence length of 200 characters and a learning rate of $2 \times 10^{-5}$. To prevent overfitting and ensure the model generalized well, an *Early Stopping* mechanism was implemented. This technique effectively concludes the training process when performance on a validation set ceases to improve [28,29]. The performance of the resulting fine-tuned models was then comprehensively evaluated on the 30% held-out test set using Python's scikit-learn library to calculate key performance metrics including accuracy, precision, recall, and F1-score. These evaluation results are detailed in Tables 1 and Table 2.

**Thai Model Performance**

As shown in Table 1, the Thai BERT model demonstrated exceptionally high performance for two of the three categories.

The "Sharing experience" class achieved an *F1-score* of 0.95, with relatively strong *Precision* (0.97) and strong *Recall* (0.94), suggesting that the model captures most instances of this

**Table 1**. **Performance metrics for the fine-tuned BERT model on the Thai test set.** Overall accuracy: 96.24%.

| Category | Precision | Recall | F1-score |
|---|---|---|---|
| Sharing experience | 0.97 | 0.94 | 0.95 |
| Inquiring | 0.40 | 0.40 | 0.40 |
| Non-informative | 0.98 | 0.99 | 0.98 |

**Table 2**. **Performance metrics for the fine-tuned BERT model on the English test set.** Overall accuracy: 80.89%.

| Category | Precision | Recall | F1-score |
|---|---|---|---|
| Sharing experience | 0.75 | 0.82 | 0.78 |
| Inquiring | 0.53 | 0.60 | 0.56 |
| Non-informative | 0.88 | 0.82 | 0.85 |

class while maintaining reasonable accuracy. The "Non-informative" class performed the best overall, with an *F1-score* of 0.98, benefiting from near-perfect *Precision* (0.98) and high *Recall* (0.99).

Conversely, the "Inquiring" class proved to be the most significant challenge, with a low *F1-score* of 0.40. This score reveals a substantial weakness in the model's ability to handle questions.

Low *Recall* (0.40): This is the most critical issue. It signifies that the model correctly identified only 40% of all actual "Inquiring" comments in the test set. The confusion matrix would show that the remaining 60% (False Negatives) were predominantly misclassified as "Sharing experience" or "Non-informative". This suggests a significant linguistic ambiguity where questions in Thai social media are often embedded within narrative text, making them difficult for the model to distinguish.

Low *Precision* (0.40): This indicates that when the model predicted a comment was "Inquiring", it was correct only 40% of the time. This is due to a high number of False Positives, where comments from the other two classes were incorrectly labeled as "Inquiring".

### English Model Performance

The English BERT model exhibited a similar performance pattern, though the challenge with the "Inquiring" class was less severe (Table 2).

The model performed well on the Sharing experience" and Non-informative" classes, achieving *F1-scores* of 0.78 and 0.85, respectively. This indicates robust and reliable classification performance for the majority of comments, with particularly strong *Precision* on the "Non-informative" class (0.88).

The "Inquiring" class remained the weakest, with an *F1-score* of 0.56.

It's moderate *Recall* (0.60): The model successfully identified 60% of all true "Inquiring" comments, a considerable improvement over the Thai model. However, the confusion matrix would still show a notable number of False Negatives, where nearly 40% of questions were missed.

Moderate *Precision* (0.53): With a *Precision* of 0.53, about half of the comments that the model labeled as "Inquiring" were indeed questions. The other half were False Positives, indicating that the model still struggles with comments that have question-like features but are not explicit inquiries. The primary source of confusion was with the "Sharing experience" class, where comments with question-like features were often misclassified.

In summary, the fine-tuned BERT model demonstrated strong overall performance across both test sets, achieving high accuracy in classifying "Sharing experience" and "Non-informative" categories, with consistently elevated *precision*, *recall*, and *F1-scores*. However, performance was notably lower for the "Inquiring" category, with comparatively reduced *precision*, *recall*, and *F1-scores*, indicating that this class remains more challenging for the model to predict accurately. These results suggest that while the model is effective in identifying the majority of content types, further refinement is required to improve its ability to recognize inquiring statements.

**Large language model (LLM) for summarization.** We integrate generative AI for comment summarization, employing the Google Gemini 2.0 Flash model through LangChain [30]. We utilize LangChain as our Generative AI framework, which facilitates calls to the Google Gemini API and structures summarization as desired by the user. The Google Gemini 2.0 Flash model serves as the model for summarizing sentiment, employing the Retrieval-Augmented Generation (RAG) approach by leveraging data from a vector database. These vectors are generated by embedding all user-desired comment texts. This RAG

approach is applied to ensure domain-specific accuracy and prevent the model from generating fabricated information (i.e., hallucinations) within the generative AI component of the NCDs Listener system. The question-answering capability of generative AI is based on a predefined set of questions to guide Gemini in generating summaries. The primary objective of these LLM-generated summaries is to enable users without expertise in graph analysis or data analytics to comprehend content related to non-communicable diseases (NCDs) within online social media.

Prompt template and predefined questions: The prompt template serves as a structured framework to guide the responses of generative AI. In this context, generative AI addresses inquiries based on the aggregated comments and responds in English. Pre-defined questions are designed to align with user requirements. To determine the preferred summary approach among users, 2 summary approaches have been defined to identify which method best aligns with user preferences. These approaches are as follows:

**General Response:**

*Prompt: "Can you summarize the comments? What are the characteristics of the comments?"*
Example of results:

> "The majority of comments express concern and provide encouragement for lung cancer patients. These comments emphasize support and highlight various perceived causes of the disease, including smoking, air pollution, and other uncertain factors, reflecting both the unpredictability of life and advancements in medical science. This General response aims to foster an online community that promotes mutual support and information sharing in an encouraging and compassionate manner."

**More Medical Focus Response:**

*Prompt: "Can you summarize the comments? What are the diseases? What are the symptoms of the disease? And other interesting information?"*
Example of the results:

> "An analysis of the comments reveals that many users provide encouragement to lung cancer patients. While specific symptoms of the disease are not explicitly mentioned, references to treatment methods such as chemotherapy and radiation therapy are present, with some patients identified as being in stage 4. Additionally, various potential causes of lung cancer are discussed, including air pollution, cigarette smoke, dietary factors, and even the COVID-19 vaccine, though the latter is presented as an assumption. Notably, some comments indicate that individuals who have never smoked can still develop lung cancer. The discussion also reflects a broader perspective that cancer can affect anyone, regardless of smoking history. Furthermore, there is a strong presence of messages conveying hope and optimism, attributing it to advancements in modern medicine."

## Dashboard design process

In designing the dashboard, a review of previous research, as discussed [3–13], was conducted. The findings indicated that various studies utilized different types of charts, such as bar charts, pie charts, and line charts, even when presenting the same variables. To ensure that the dashboard design aligns with user needs, an assessment of required variables and the appropriateness of chart types was conducted. This assessment involved a total of 10 participants, comprising 5 general users or students from the Faculty of Science, Department of Statistics, Khon Kaen University who had not yet completed the introductory data science

course, and 5 data scientists or students from the same department who had completed the course and had shown a high level of proficiency in the subject matter, along with faculty members from the Department of Statistics, Khon Kaen University for the academic year 2024. The research process was carried out as follows:

1. The researcher presented all extracted variables to the participants, with each variable represented using 4 different chart types.
2. The participants subsequently completed an assessment consisting of 8 items to evaluate the relevance of each variable and the appropriateness of the chart types. Additionally, interviews were conducted to gather further feedback and identify areas for improvement.
3. The researcher analyzed participants' preferences regarding the variables and chart types using the mean score and frequency of selection.
4. Based on the findings, the researchers developed the NCDs Listener dashboard. The dashboard design process is presented in Fig 3.

The study identified a total of 8 variables that could be extracted and visualized:

1. Gender of the patient
2. Source of the opinion (Whose experience is being shared?)
3. Categorized comments (From machine learning models)
4. Mentioned diseases
5. Mentioned symptoms
6. Mentioned treatment
7. Mentioned causes of the disease or patient behaviors
8. Most frequently used words across all opinions

For each variable, participants were asked 3 key questions:

1. Do you consider this variable to be relevant and capable of providing useful information? (Likert scale [31])
2. Which chart type do you find most suitable for representing this variable? (Selection 1 chart from 4 options, including bar charts and pie charts designed by the researchers)
3. Do you have any suggestions or recommendations for improvement? (Open-ended response)

The findings will be utilized to develop charts and dashboards for the NCDs Listener tool, which will subsequently be tested with users in the following section.

### Evaluation of dashboard communication effectiveness and selection of generative AI-based summarization approaches

This section presents an experimental study in which target users interact with the results of 6 case studies generated by the NCDs Listener tool. Participants will review both the dashboard visualizations and generative AI-based summaries. The experiment consists of 3 key components: experiment design, question formulation, evaluation process, and data analysis.

**Design of the assessment for ability and satisfaction.** This study evaluates user satisfaction and comprehension of the dashboard, as well as the selection of the most effective generative AI-based data summarization method within the NCDs Listener tool. A mixed

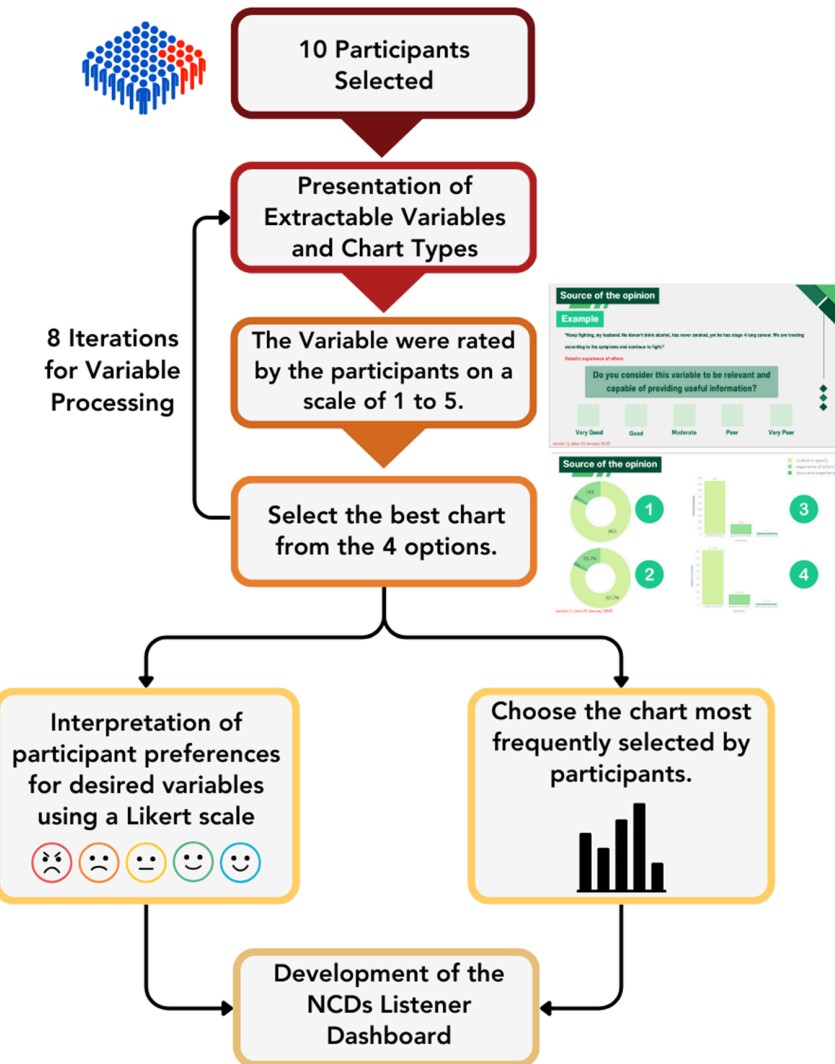

**Fig 3. The dashboard design process.** The process began with the selection of 10 participants. Each was presented with a set of extractable variables and four chart types for visualization. For every variable, participants rated its importance on a 1–5 Likert scale and selected the best chart from the four options presented. This was repeated for eight variables. Ratings were analyzed to interpret preferences for desired variables, while chart type preferences were determined by identifying the most frequently selected option. The results of these evaluations were used to guide the presentation and design of the final visualizations. These visualizations were then integrated into the development of the NCDs Listener dashboard, ensuring that both content and presentation reflected participant preferences.

research design is employed, combining a cross-sectional study and an experimental research approach. The study population is divided into 2 groups:

1. General users, including students from the Department of Statistics, Faculty of Science, Khon Kaen University who have not yet completed the introductory data science course in the academic year 2024.
2. Data scientists, including students from the Department of Statistics, Faculty of Science, Khon Kaen University, who have completed the introductory data science course in the

academic year 2024, as well as faculty members from the Department of Statistics in the same academic year.

The study involves a total of 30 volunteers. This cohort is distinct from the 10 volunteers who participated in the initial dashboard design interviews. The 30 participants provided an actual power of 81% (as detailed in Table 3). were selected through a combination of stratified sampling and simple random sampling from the list of students in the Department of Statistics, Faculty of Science, Khon Kaen University. The sample size was determined using the G*Power program, following Cohen's [32] effect size criteria. The calculation was based on an effect size of 0.80, with a 90% confidence level and an error margin of 0.10. Although increasing sample size generally improves accuracy, the chosen sample size was justified by the actual power of 0.81. This demonstrates that the sample is interpretable and adequately powered for the study's aims. The results of the sample size calculation are presented in the following Table 3.

Participants were exclusively drawn from undergraduate statistics students to ensure sufficient data literacy for meaningful interaction with both dashboard visualizations and generative AI summaries [33]. Restricting the sample to a single department also enabled stratification by prior Data Science coursework, thereby controlling between-subject variability and facilitating direct comparisons across technical proficiency levels [34]. Moreover, the modest cohort size reflects the practical constraints of time, budget, and logistics inherent to early-stage prototype testing [35]. The participant groups for evaluating the dashboard and generative AI's communication effectiveness were not the same individuals who participated in the prior identification of desired variables and charts.

We obtained written informed consent from all participants before data collection. Sample collection and assessments were conducted between February 27, 2025, and March 10, 2025.

**Evaluation process.** Before data collection, we will provide participants with an overview of the research, including its objectives and the results generated by the NCDs Listener web application. This information will be presented through slide presentations and oral explanations. The data collection process consists of the following steps:

A. Assessment of dashboard communication ability

1. Participants will be randomly assigned to view the NCDs Listener tool results presented across 6 dashboards.

**Table 3. Parameters for sample size calculation in evaluating dashboard communication and generative AI summaries.**

| t tests - Means: Difference between two independent means | | |
|---|---|---|
| **Analysis:** | **A priori: Compute required sample size** | |
| **Input:** | **Tail(s)** | One |
| | **Effect size $d$** | = 0.80 |
| | **err prob ($\alpha$)** | = 0.10 |
| | **Power ($1 - \beta$ err prob)** | = 0.80 |
| | **Allocation ratio $N_2/N_1$** | = 1.00 |
| **Output:** | **Noncentrality parameter $\delta$** | = 2.19 |
| | **Critical $t$** | = 1.31 |
| | **Df** | = 28 |
| | **Sample size group 1** | = 15 |
| | **Sample size group 2** | = 15 |
| | **Actual power** | = 0.81 |

2. They will complete 2 assessment tasks designed to evaluate the effectiveness of dashboard communication.
3. Following the assessment, participants will be interviewed to provide additional suggestions or identify areas for improvement.

B. Assessment of generative AI-based opinion summaries

1. Participants will be randomly assigned to review data and summaries generated by generative AI, categorized into 2 summary approaches: more medical focus response, general response
2. Each summary approach consists of 6 types, resulting in a total of 12 summaries. Each participant will review all 12 summaries.
3. Participants will complete 3 evaluation tasks related to generative AI-based summaries.
4. An interview will follow to gather additional suggestions or areas for improvement.

The entire evaluation process, encompassing both the assessment of dashboard communication ability and the assessment of generative AI-based opinion summaries, is illustrated in Fig 4.

Finally, the collected data will be analyzed using statistical methods, ensuring rigorous evaluation and interpretation of the results.

**Question formulation.** The question formulation is structured into 2 main components: (1) the evaluation of dashboard communication effectiveness and (2) the overall assessment of generative AI's summarization of user comments.

(1) The evaluation of dashboard communication effectiveness

For each case study, 5 questions are used for evaluation, including 2 questions specifically designed to assess the effectiveness of the dashboard. These questions are formulated based on 2 key aspects:

1. Appropriateness of the amount of information for task performance
2. Quality of the information provided

The specific questions used for dashboard evaluation are as follows:

A: Does the dashboard provide relevant information regarding the post on [specified disease]?

B: Does the dashboard enhance your understanding of the current situation of [specified disease] disease? (Responses are on the Likert scale.)

(2) The overall assessment of generative AI's summarization of user comments.

For each case study, 5 evaluation questions are used, including 3 questions specifically designed to assess the performance of generative AI. These questions are formulated based on 3 key evaluation criteria:

1. Reasoning – The AI's ability to generate logical and coherent summaries.
2. Societal Impact – The AI's capability to capture the broader social implications of the disease.
3. Domain Knowledge – The AI's understanding of the representation of medical information.

The specific questions used in the assessment are as follows:

A. How effectively does this sentence communicate information about [specified disease] to the general public?

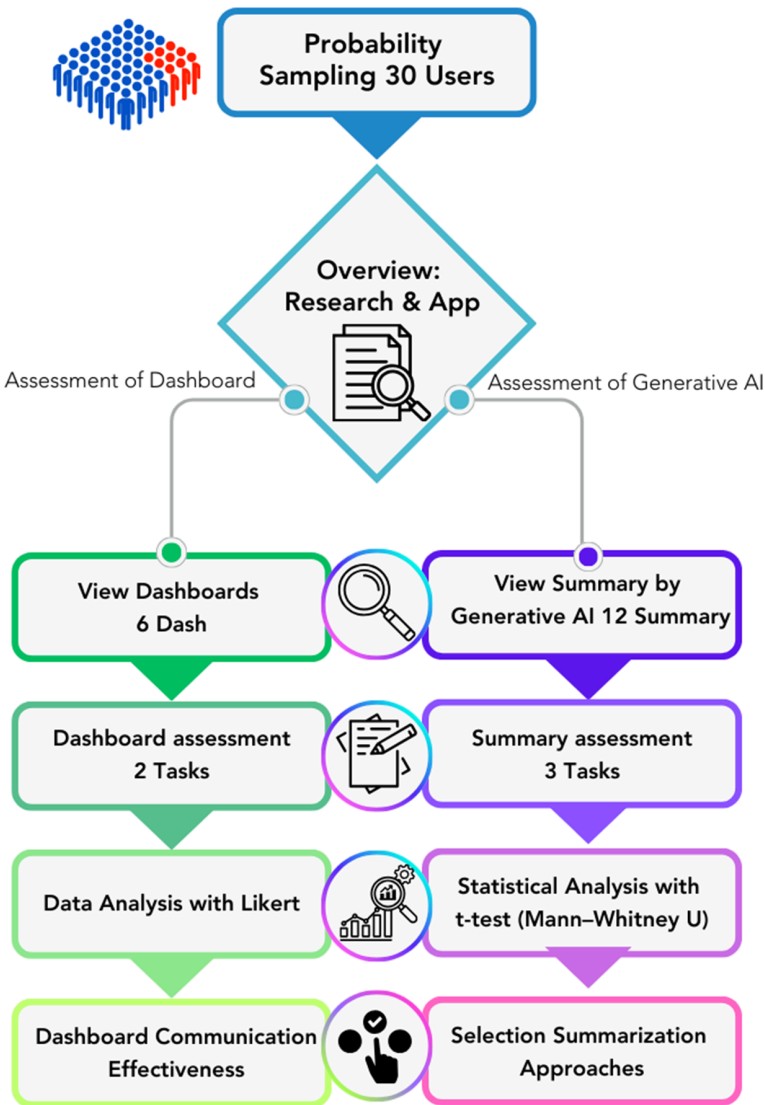

**Fig 4. Evaluation strategy illustrates the process for assessing both the communicative effectiveness of the dashboard and the optimal summarization approach for generative AI content.** A probability sample of 30 users was selected and divided into two main assessment paths. The first path involved evaluating six dashboards through two assigned tasks, followed by data analysis using a Likert scale to measure user perceptions, ultimately determining the dashboard's communication effectiveness. The second path focused on assessing generative AI summaries, where participants reviewed twelve AI-generated summaries and completed three evaluation tasks. Statistical comparisons were conducted using the t-test and Mann–Whitney U test to identify significant differences in performance and preferences. The combined results from both assessments informed the selection of the most suitable summarization approaches and guided improvements in dashboard presentation and AI integration.

 B. To what extent does this sentence demonstrate an understanding of the societal impact of [specified disease], as perceived from the perspective of the media?

 C. How reasonable do you find this summary of opinions? (Responses are on the Likert scale.)

 D. Additional suggestions or feedback (Open-ended response.)

Where [specified disease] refers to 1 of the 6 predefined case studies:

1. Cancer – as presented on a news page [36].
2. Cancer – as shared by cancer patients [37].
3. Heart disease – as presented on a news page [38].
4. Heart disease – as presented on a medical page [39].
5. Diabetes –as presented on a doctor's page providing medical knowledge [40].
6. Diabetes – as presented on a doctor's page sharing experiences from treating patients [41].

**Data analysis.** The analysis of user needs and satisfaction is primarily conducted using statistical methods to ensure an objective evaluation.

(1) Assessment of dashboard communication ability

The effectiveness of dashboard communication is assessed based on Likert scale ratings, with the following interpretation criteria for the average scores, as shown in Table 4.

These statistical measures will be used to evaluate user feedback and determine the overall effectiveness of the dashboard in conveying relevant information.

(2) Statistical analysis of generative AI-based opinion summaries

To evaluate user satisfaction with generative AI-generated summaries, a Mann–Whitney U test will be conducted using the SPSS statistical software. This test will determine whether there is a significant difference in satisfaction levels between the more medical-focused response and the general response. The results of the hypothesis test will be interpreted based on the P-value. This analysis will provide insights into how different user groups perceive the effectiveness of the 2 summary approaches, supporting further refinement of the NCDs Listener tool.

## Results

This section presents the findings from the survey on user needs, satisfaction, and comprehension of the NCDs Listener tool. The data were collected from participants and were categorized into three key areas: User-Centered Dashboard Design, Evaluation of Dashboard Communication Effectiveness, and Selection of Generative AI-Based Summarization Approaches.

### User-centered dashboard design

The researchers conducted a user needs assessment to guide the development of the dashboard. Participants were asked to select their preferred variables from a predefined set of 8 variables and to identify the most suitable chart type for each selected variable. The findings from this assessment are summarized in the following Table 5.

**Table 4. Likert scale interpretation thresholds for user-desired variables and dashboard communication effectiveness.**

| Mean score | Interpretation |
|---|---|
| 4.21 - 5.00 | Very Good |
| 3.41 - 4.20 | Good |
| 2.61 - 3.40 | Moderate |
| 1.81 - 2.60 | Poor |
| 1.00 - 1.80 | Very Poor |

**Table 5. Preferred variables (mean score ≥ 4.21) and chart types for dashboard construction based on user assessment (n=10).**

| Variable | Mean | Interpretation | Chart type |
|---|---|---|---|
| Gender of the patient | 4.67 | Appropriate | pie |
| Source of the opinion | 3.56 | Not appropriate | pie |
| Categorized comments | 3.56 | Not appropriate | pie |
| Mentioned diseases | 5.00 | Appropriate | bar |
| Mentioned symptoms | 4.67 | Appropriate | bar |
| Mentioned treatment | 4.89 | Appropriate | bar |
| patient behaviors | 4.78 | Appropriate | bar |
| Most frequent words (all opinions) | 3.33 | Not appropriate | bar |

Based on the results presented in Table 5, the researchers identified variables with an average score exceeding 4.21 as those that users preferred to be displayed on the dashboard. Specifically, the variable "Gender of the patient" will be visualized using a pie chart, while "Mentioned diseases", "Mentioned symptoms", "Mentioned treatment", and "Patient behaviors" will be presented using a bar chart. The remaining variables were categorized by the researchers as hidden variables, which users can access at any time if desired.

## Evaluation of dashboard communication effectiveness

The researchers conducted an assessment of satisfaction and understanding regarding the dashboard by having users review the results of the NCDs Listener tool across all 6 cases, as presented in Table 6.

The results presented in Table 6 indicate that both data scientists and general users assessed the dashboard as having good communication ability across all case studies. However, the data scientists group consistently provided lower average scores than the general users group for each evaluation criterion. It can be concluded that the general user group tended to perceive the dashboard as having higher data quantity and quality compared to the data scientist group.

## Selection of generative AI-based summarization approaches

In this section, we evaluate the communication capabilities of generative AI to identify the optimal summary approach preferred by both user groups, categorized by each criterion based on the questions used.

The analysis, as presented in Table 7, revealed a statistically significant difference (P<0.001) in the evaluation scores of generative AI summary comprehension. This difference was

**Table 6. Dashboard communication effectiveness evaluation results for data scientists and the general public (n=30).**

| Case Studies | General Public (n=15) | | Data Scientists (n=15) | |
|---|---|---|---|---|
| | Mean | Interpretation | Mean | Interpretation |
| 1 | 3.88 | Good | 3.58 | Good |
| 2 | 3.97 | Good | 3.47 | Good |
| 3 | 4.00 | Good | 3.63 | Good |
| 4 | 3.97 | Good | 3.70 | Good |
| 5 | 4.28 | Very Good | 3.73 | Good |
| 6 | 4.25 | Very Good | 3.73 | Good |
| ALL | 4.05 | Good | 3.65 | Good |

**Table 7. User preference for generative AI summarization: Medical-focused vs. general responses (n=30).**

| Evaluation criteria | Approach | Test Statistic | P-value | Mean Rank |
|---|---|---|---|---|
| Reasoning | General | | | 221.13 |
| | Medical | 12935.50 | $8.65 \times 10^{-8}$[a] | 163.87 |
| Societal Impact | General | | | 215.21 |
| | Medical | 14071.00 | $2.00 \times 10^{-5}$[a] | 169.79 |
| Domain Knowledge | General | | | 218.29 |
| | Medical | 13479.50 | $1.00 \times 10^{-6}$[a] | 166.71 |

[a] indicates p-value < 0.001

observed between users who read summaries generated with a medical-focused response and those with a general response across all evaluation criteria: Domain Knowledge, Societal Impact, and Reasoning. Users exposed to the medical-focused response summaries consistently exhibited a significantly higher Mean Rank. This finding suggests that users who read the more medically focused summaries demonstrated a superior understanding of the generative AI summaries developed by the researchers in all evaluated aspects. Consequently, the medical-focused response was adopted for generating summaries within the NCDs Listener tool, effectively addressing users' specific needs.

## Web application development

In this section, the researcher utilized the results to develop the NCDs Listener tool, ensuring alignment with user requirements. The dashboard was designed to display the variables specified by the user while allowing access to additional variables at any time. Users can modify the data according to their needs, with adjustments dynamically updating the charts and data presented in the table above. Additionally, users can export the table data in CSV format for future use. At the bottom of the dashboard, a summary is generated by generative AI using the more medical-focused response, as illustrated in Fig 5.

## Performance and functional comparison

In this section, we tested our tool with real users, from the data extraction process to image display, using posts containing 30 to 2,000 comments from both Facebook and Reddit. We found that processing on Facebook took an average of 5 minutes, with a maximum of 80 minutes. For Reddit, the average processing time was 1 minute, with a maximum of 30 minutes. This difference in processing time is attributed to the complexity of the websites; Facebook requires the bot to navigate several additional steps for comment access before data collection, whereas Reddit involves only one to two steps. Furthermore, the speed of Thai word segmentation is slower than that of English.

For comparing functional differences between our tool and existing ones, we conducted in-depth research, including free tool trials, reviews, and pricing documentation [42–45]. We observed that most existing tools primarily focus on posts or search terms. While they offer continuous updates, provide demographic data (such as gender, likes, posting time, and hashtags), sentiment analysis, topic analysis, and data insights—functions largely similar to ours—these tools typically require a paid subscription. Our tool highlights methodologies for web data ingestion, utilizing technologies such as Selenium and Beautiful Soup to facilitate the aggregation of post comments without requiring programming. Furthermore, we implemented keyword matching and a BERT Comment classification model for the cleaning and

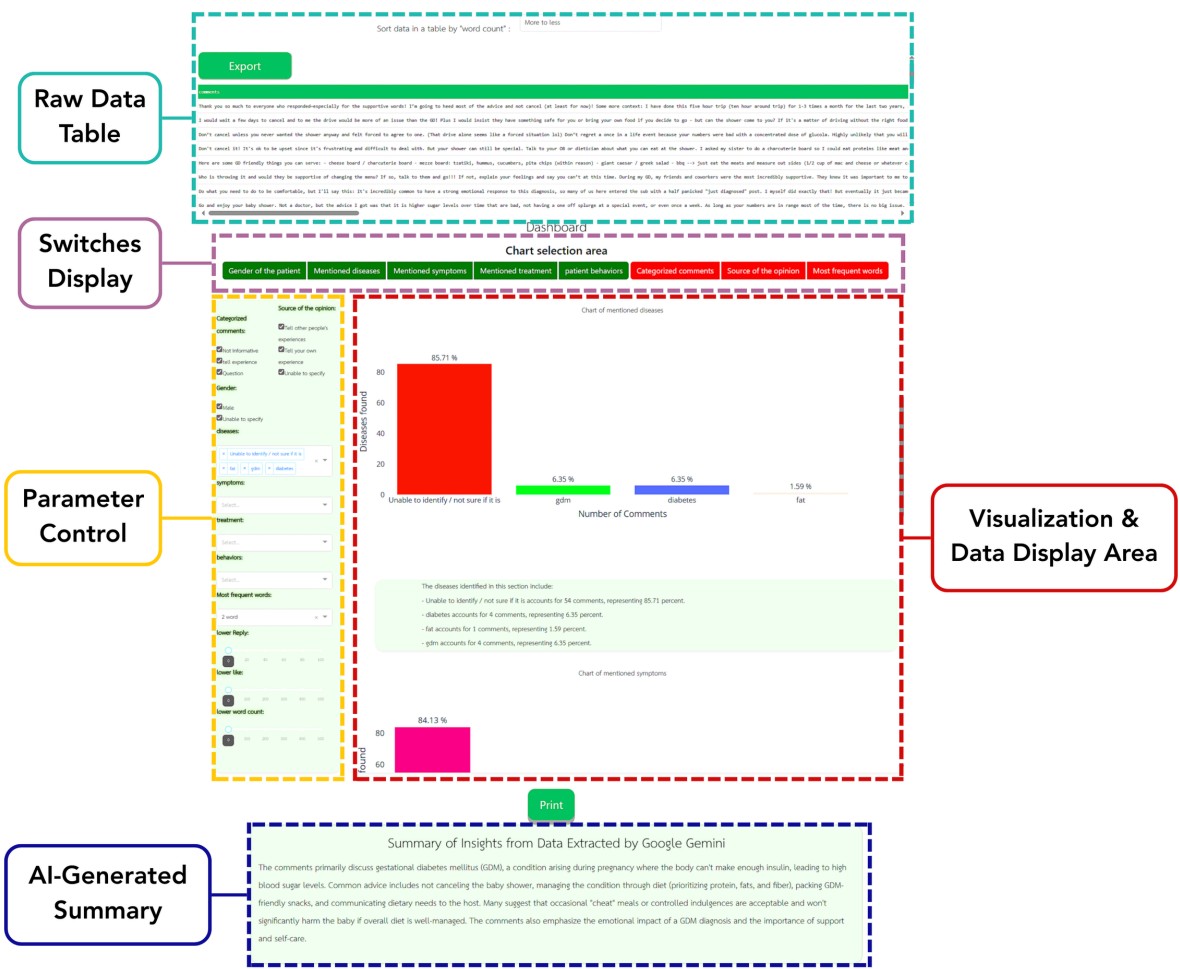

**Fig 5. NCDs listener example dashboard interface displaying comprehensive analytical results and data visualization capabilities.**
The dashboard features an editable and exportable raw data table at the top, enabling direct data manipulation and export capabilities. The left panel presents a customizable dashboard area with interactive toggle switches displayed in green (active/enabled) and red (inactive/disabled) states for dynamic parameter control. The central visualization area displays statistical charts and graphical representations of the processed NCD data. The bottom of the chart contains a legend and a data display area for a better understanding of the data. At the bottom, an AI-generated summary section provides automated insights from Google Gemini integration, offering intelligent analysis and interpretation of the uploaded content. This integrated interface facilitates comprehensive non-communicable disease data analysis through intuitive visualization, real-time parameter adjustment, and automated analytical summaries for enhanced research workflow efficiency.

analysis of Non-Communicable Diseases (NCD)-related data. Generative AI technologies are also integrated to enable deeper insights.

## Ethical considerations in social media data utilization

Ethical considerations are paramount when using social media data, particularly concerning personal identifiers, comments, and other potentially sensitive information. Legally, Article 9(2)(e) of the General Data Protection Regulation (GDPR) addresses the processing of special categories of personal data that the data owner has publicly made available, which could potentially be used to identify an individual [46].

To protect both users and those whose data are collected, the researchers removed names after cleaning the data. This measure helps prevent the identification of individuals, allowing for valuable research insights from publicly available social media discussions while safeguarding privacy.

## Limitations

This research focuses on the development of a social media listening tool for Non-Communicable Diseases (NCDs), designed for the general public and data scientists via a web application. The primary goal was to create a tool that meets user needs and provides understandable insights, prioritizing both speed of development and methodological rigor. The user-centered development process involved multiple phases, including an initial dashboard design phase with 10 participants, followed by a testing phase with 30 participants to evaluate dashboard communication effectiveness and select a generative AI-based summarization approach. This study utilized a relatively small dataset from 30 participants in the evaluation phase, all of whom were statistics students from a single institution. While this homogeneous cohort was beneficial for controlled initial usability testing and yielded statistically significant preliminary findings, with a calculated actual power of 81% (as detailed in Table 3), the dataset is too limited in size and diversity to support broad generalizations. The current results should therefore be interpreted strictly as indicative of initial system performance under controlled conditions, rather than as definitive evidence of its effectiveness across diverse populations.

Currently, our tool supports content in both Thai and English and is designed to extract public comments exclusively from two social media platforms: Reddit and Facebook. The tool demonstrates optimal effectiveness when applied to comment data related to Non-Communicable Diseases (NCDs). Furthermore, as our web application processes comments and chat-like messages, it encounters inherent variations in language, writing style, and terminology. The nature of discussions within a post also influences the results. While this approach allowed us to develop the tool effectively within our limitations, it is important to note that these factors may lead to some variations in the accuracy of the results. Additionally, the input of unreliable data by users can impact the results, even with the assistance of our BERT model designed to filter uninformative content.

## Discussion and conclusion

The NCDs Listener is a tool designed to monitor and analyze social media data related to Non-Communicable Diseases (NCDs). The system collects data from platforms such as Facebook and Reddit through specified URLs. Subsequently, it enhances data quality using techniques such as list collapsing and Natural Language Processing (NLP), including tokenization, stopword removal, and other preprocessing methods. To extract meaningful insights, the system employs knowledge extraction techniques that utilize keyword matching and topic classification, leveraging the Bidirectional Encoder Representations from Transformers (BERT) model. The BERT model employed in this study has been trained on over 3,000 datasets and demonstrates high accuracy, exceeding 80% for both Thai and English data; however, it still presents limitations in classifying text in the "Inquiring" category. Finally, the processed information is presented via an interactive dashboard with AI-generated commentary. This provides users with comprehensive and accessible insights to facilitate a deeper understanding of NCD-related discussions on social media.

This application serves as a valuable tool for researchers investigating the social aspects of diseases of interest, consistent with the objectives found in the existing literature we have

studied [5–10,12]. Concurrently, our tool differs from existing solutions primarily focused on marketing [3,4].

For this tool, a knowledge extraction approach combining keyword matching with BERT was employed to identify important data features. This approach differs from the existing literature we have studied, which either manually performed knowledge extraction or utilized pre-trained models solely for analyzing emotional responses in comments [5–11,13]. Additionally, our tool incorporates data visualization to present findings, which aligns with the presentation methods used in the existing literature we have studied [5,7,9–13]. However, our tool augments this presentation by summarizing overall comments using Generative AI.

The criteria used to measure the communication capabilities of both the developed Dashboard and the Generative AI align with established research findings on effective communication assessment [14,15].

The primary distinction between the previous research [16]and this current study lies in its responsiveness to user needs. The original work did not incorporate user needs assessment or evaluate the communication effectiveness of the presentation media. Consequently, this research focuses on developing the existing tool to align with user requirements and measuring the communication efficiency of both the dashboard and the Generative AI summaries produced by our tool.

To better align with user needs, we developed this tool by soliciting user preferences for specific variables and chart displays. The resulting dashboard presents a total of 6 variables: the patient's gender (4.67), the disease mentioned (5.00), the symptoms mentioned (4.67), the treatment method mentioned (4.89), and the patient's behavior mentioned (4.78). These variables were identified based on a user needs survey, with all variables receiving an average score exceeding 4.21. The most suitable variable for a pie chart is the patient's gender, while the remaining five variables are best visualized using a bar chart. Any variable scoring below 4.21 remains hidden by default but can be accessed by users if needed.

Once the desired dashboard configuration is obtained, an assessment of its communication capability is necessary. The findings indicate that the overall dashboard demonstrates good communication capabilities between the two sample groups. However, the data scientist group assigned a lower average score (3.64) compared to the general user group (4.06), suggesting that the dashboards generated provide a greater quantity and quality of information to the general public than to data scientists.

Based on the test results of the Selection of generative AI-based summarization Approaches, users rated the more medical-focused responses' understanding of Domain Knowledge (221.13), Societal Impact (215.21), and Reasoning (218.29) as superior to the general responses' understanding across all aspects. Consequently, the researchers opted to utilize generative AI to generate summaries based on the more medical-focused response within the NCDs Listener tool, effectively addressing the specific needs of the users.

Several promising directions for future research are identified to support the continued advancement of the proposed tool. Three key areas have been recognized as opportunities to improve the system's performance, scalability, and applicability in broader research contexts. First, regarding data collection, the current tool is limited to retrieving data from Facebook and Reddit using URLs. Future development should incorporate keyword-based search functionality to facilitate more efficient data gathering for users and expand the tool's coverage to include other social media platforms such as X (formerly Twitter) and Telegram.

Following the first point, since our system relies on URLs from social media platforms that are frequently updated, our web scraping tools must also be continuously adapted. This is particularly critical because social media platforms, especially Facebook, frequently update their HTML tags and implement protective measures against scraping, leading to constant changes

in their HTML structure. Therefore, we propose integrating AI to enhance the web scraping process by enabling the AI to autonomously understand the components of the user interface and identify relevant content areas. This approach eliminates the need for constant manual updates to the scraping code.

In terms of knowledge extraction, the current tool utilizes keyword matching and machine learning techniques, which can limit the scope of identifying relevant information related to Non-Communicable Diseases. This limitation arises due to variations in language use among individuals and the time-consuming nature of the current approach. To address these challenges, future work may explore the integration of semantic relationship models or large language models (LLMs) to extract additional variables and reduce current limitations.

In addition to these primary directions, there are further suggestions for future improvement. These include expanding the user base to target groups such as patients and medical professionals, who are more likely to seek information related to communities of individuals with Non-Communicable Diseases. Moreover, enhancing the overall user interface of the tool to make it more modern and user-friendly is also recommended.

The present evaluation provides a statistically significant preliminary validation of the tool's usability and communication effectiveness within a controlled academic setting. However, given the limited sample size and the specific background of participants, these results cannot be generalized to the wider population. Future development should therefore be centered on a rigorous re-examination of user needs through expanded evaluation that specifically includes key end-user groups: patients and frontline healthcare professionals. This expanded validation will employ rigorous sampling strategies and comprehensive evaluation methodology that assesses not only technical performance metrics (e.g., processing speed, functional efficiency) but also user-centric factors such as interface aesthetics, tool reliability, and overall usability. The insights gained from these diverse user groups will be instrumental in developing a strategic implementation roadmap for the tool's adoption in real-world clinical and public health contexts, enabling robust generalization of findings beyond the current controlled academic environment.

In terms of artificial intelligence models for insight extraction, several key areas warrant further investigation. For machine learning-based topic classification, future research should focus on enhancing the model's ability to distinguish between different question types through architectural improvements and expansion of the test dataset to achieve greater statistical power. Critically, bias reduction in data classification represents a priority area, necessitating a transition from human-annotated opinion characteristics to automated machine learning or deep learning approaches for feature identification and classification.

Regarding large language model development, the incorporation of additional performance metrics beyond current measures is essential for comprehensive evaluation. While standard text generation metrics such as ROUGE and BLEU scores are widely used in natural language processing, there are currently no established gold standards specifically designed for evaluating social media-derived non-communicable disease (NCD) data summaries. This limitation extends even to the application of conventional metrics like ROUGE and BLEU scores, as appropriate reference standards for our specific domain remain undefined. Future implementations should therefore focus on developing domain-specific evaluation frameworks that incorporate adapted versions of ROUGE and BLEU scores alongside medical-entity coverage metrics, particularly as the target population expands to include both patients and healthcare providers. A critical priority for future work will be establishing gold standards tailored to social media NCD analytics, which will enable the creation of robust evaluation methodologies specific to our application domain. These enhancements will enable more comprehensive and contextually appropriate evaluation

frameworks, supporting the model's applicability and validation across diverse clinical contexts.

Most current health data analysis still depends on traditional methods. Although some open-source platforms do not exactly, they are often not fully suitable for comprehensive health analysis. Certain platforms use AI-powered, NLP-enabled large language models (LLMs), but these typically rely on a single model architecture, which limits their flexibility and depth. In this paper, we demonstrate the development and integration of multiple NLP and AI techniques to overcome these limitations.

Although social media data has the potential to address some shortcomings of traditional approaches, there remains a significant lack of dedicated social media listening tools capable of conducting in-depth health data analysis. To address this gap, our tool enhances the use of social media data for comprehensive health analysis, providing insights that are easily summarized or applied across various domains without requiring extensive manual effort from researchers for data collection, analysis, and presentation.

## Author contributions

**Conceptualization:** Ratchanont Thippimanporn, Thanapong Intharah.

**Data curation:** Wuttichai Khamna, Kannika Wiratchawa.

**Formal analysis:** Ratchanont Thippimanporn, Wuttichai Khamna.

**Investigation:** Ratchanont Thippimanporn, Wuttichai Khamna.

**Methodology:** Ratchanont Thippimanporn, Wuttichai Khamna.

**Project administration:** Kannika Wiratchawa.

**Supervision:** Thanapong Intharah.

**Validation:** Wuttichai Khamna, Thanapong Intharah.

**Visualization:** Ratchanont Thippimanporn, Wuttichai Khamna.

**Writing – original draft:** Ratchanont Thippimanporn, Wuttichai Khamna.

**Writing – review & editing:** Kannika Wiratchawa, Thanapong Intharah.

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
