## [Decision Letter · Decision Letter 0]

26 May 2025

PONE-D-25-21639Democratizing Social Media for Health Research: LLM-Powered Data Analytics Platform for NCDsPLOS ONE

Dear Dr. Intharah,

Thank you for submitting your manuscript to PLOS ONE. After careful consideration, we feel that it has merit but does not fully meet PLOS ONE’s publication criteria as it currently stands. Therefore, we invite you to submit a revised version of the manuscript that addresses the points raised during the review process.

As you can see, Reviewer 1 has concerns about technical depth and evaluation rigor, while Reviewer 2 raised concerns about the manuscript structure. Kindly ensure all figures and tables are embedded within the main text. Provide a clear and reasonable justification if any comment cannot be addressed. Responses should follow the journal’s guidelines and be submitted in a separate supplementary file, with edits highlighted in yellow. Importantly, to comply with PLOS ONE’s Data Availability Policy, the authors must make all datasets used for pretraining and fine-tuning publicly accessible. 

We look forward to receiving your revised manuscript.

Kind regards,

Issa Atoum

Academic Editor

PLOS ONE

Journal Requirements:

4. We note you have included a table to which you do not refer in the text of your manuscript. Please ensure that you refer to Table 5 in your text; if accepted, production will need this reference to link the reader to the Table.

Reviewers' comments:

Reviewer's Responses to Questions

**Comments to the Author**

1. Is the manuscript technically sound, and do the data support the conclusions?

Reviewer #1: Yes

Reviewer #2: Yes

2. Has the statistical analysis been performed appropriately and rigorously? 

Reviewer #1: Yes

Reviewer #2: Yes

3. Have the authors made all data underlying the findings in their manuscript fully available?

Reviewer #1: Yes

Reviewer #2: Yes

4. Is the manuscript presented in an intelligible fashion and written in standard English?

Reviewer #1: Yes

Reviewer #2: No

5. Review Comments to the Author

Reviewer #1: Thank you for submitting your manuscript on the NCDs Listener platform. I found the motivation behind developing an open-source social listening tool for health research to be compelling and timely. However, I have several concerns that need to be addressed before the manuscript can be considered for publication.

Major Concerns:

1. System Architecture and Technical Details

The current presentation of your system design, particularly Figure 1, lacks the technical depth expected for a paper of this nature. The diagram shows only high-level components without revealing the underlying architecture. For a platform described as "LLM-powered," readers need to understand:

- The specific frameworks and technologies employed

- How different components interact

- The data flow from social media platforms through your processing pipeline

- Integration details between BERT classification and LLM summarization

I suggest developing a comprehensive system architecture diagram that provides these essential details. This is particularly important for an open-source tool that others might want to replicate or extend.

2. Evaluation Methodology

While I appreciate the user study conducted, the current evaluation has limitations:

- The sample size of 30 participants is rather small for drawing robust conclusions

- The lack of comparison with existing tools (even if they are commercial) leaves questions about relative performance

- Performance metrics such as processing speed, scalability, and system reliability are not addressed

Consider expanding the evaluation to include more participants and adding comparative benchmarks where possible.

3. Technical Implementation Details

Several technical aspects require clarification:

- The BERT model training process needs more detailed documentation (dataset composition, training parameters, validation methodology)

- The integration of LangChain and Gemini 2.0 Flash should be explained in greater technical detail

- The web scraping methodology and how you handle platform-specific challenges deserve more attention

4. Research Contributions

The manuscript would benefit from a clearer articulation of its novel contributions. While making existing technologies accessible through open-source tools is valuable, the paper should better highlight what technical or methodological innovations it offers beyond implementation.

Minor Issues:

1. The abstract is somewhat lengthy and could be more concise while still conveying the key contributions.

2. Figure 4's dashboard screenshots are difficult to read. Higher resolution images would better showcase this important component.

3. The related work section could be expanded to better position your work within the existing literature.

4. Some discussion of ethical considerations regarding social media data collection would strengthen the paper.

Recommendations:

I recommend major revision for this manuscript. The core concept is sound and addresses a real need in the health research community. However, the paper requires substantial additions in terms of technical depth and evaluation rigor. Specifically:

1. Provide detailed system architecture diagrams and technical implementation details

2. Expand the evaluation with a larger sample size and consider including comparative analyses

3. More clearly articulate the research contributions and innovations

4. Address ethical considerations around social media data usage

With these revisions, this work could make a meaningful contribution to health informatics research. I encourage the authors to address these points and resubmit. The open-source nature of your tool is commendable, and with proper documentation and evaluation, it could indeed serve as a valuable resource for the research community.

Reviewer #2: The topic of the article is of high importance and it is necessary to address it.

1- Abstract should be rewritten and structured.

2- some sentences and claims do not have references, (e.g. first paragraph), please add

3-Secound paragraph is very long and should be written based on scientific writing; divided into some inter connected paragraphs

4- It is not needed to bring the Print Screen of web page in the article (Page 15). If the graphs needed to be in the text please bring the graphs separately with the reference.

5- The sampling should be mentioned: why the authors selected the samples just from students of statistics?

6- In Line 51 after the Title :Related Work , there is just one sentence and shows something is omitted or this title is redundant.

7-There is no "Methodology" part in this article, the authors should write it scientifically.

8- The title of tables is for introduction of table not for explanation, the explanation of each table should be written in detail before or after the table and mentioning the number of the given table in the text.

9- the limitations of this study should be mentioned clearly in the text.

10-The discussion section should be modified based on literature and other studies.

6. PLOS authors have the option to publish the peer review history of their article (what does this mean?). If published, this will include your full peer review and any attached files.

Reviewer #1: No

Reviewer #2: No

---

## [Decision Letter · Decision Letter 1]

13 Jul 2025

PONE-D-25-21639R1Democratizing Social Media for Health Research: LLM-Powered Data Analytics Platform for NCDsPLOS ONE

Dear Dr. Intharah,

Thank you for submitting your manuscript to PLOS ONE. After careful consideration, we feel that it has merit but does not fully meet PLOS ONE’s publication criteria as it currently stands. Therefore, we invite you to submit a revised version of the manuscript that addresses the points raised during the review process. Please submit your revised manuscript by Aug 27 2025 11:59PM. If you will need more time than this to complete your revisions, please reply to this message or contact the journal office at plosone@plos.org. Please include the following items when submitting your revised manuscript:

We look forward to receiving your revised manuscript.

Kind regards,

Issa Atoum

Academic Editor

PLOS ONE

Journal Requirements:

**Additional Editor Comments:**

While Reviewer 2 did not raise any concerns, Reviewer 1 identified several critical issues related to the system architecture depth, the comprehensiveness of the evaluation, and the clarity of the research contribution. Please ensure that all reviewer comments are thoroughly addressed in the revised manuscript. To facilitate the review process, incorporate all figures and tables directly within the manuscript. Additionally, ensure full compliance with PLOS ONE's requirements for data and software availability. This includes:

Providing a Data Availability Statement that clearly outlines how the data supporting your findings can be accessed.Sharing all relevant software and code used in the study, ideally through a public repository (e.g., GitHub, Zenodo).Ensuring that all materials, including datasets and tools, are accessible without restriction, unless ethical or legal limitations apply.

For detailed guidance, please refer to the following official PLOS ONE policies:

Materials, Software, and Code Sharing Policy (https://journals.plos.org/plosone/s/materials-and-software-sharing)Data Availability Policy(https://journals.plos.org/plosone/s/data-availability)

Reviewers' comments:

Reviewer's Responses to Questions

**Comments to the Author**

1. If the authors have adequately addressed your comments raised in a previous round of review and you feel that this manuscript is now acceptable for publication, you may indicate that here to bypass the “Comments to the Author” section, enter your conflict of interest statement in the “Confidential to Editor” section, and submit your "Accept" recommendation.

Reviewer #1: (No Response)

Reviewer #2: (No Response)

2. Is the manuscript technically sound, and do the data support the conclusions?

Reviewer #1: Partly

Reviewer #2: Yes

3. Has the statistical analysis been performed appropriately and rigorously? 

Reviewer #1: Yes

Reviewer #2: N/A

4. Have the authors made all data underlying the findings in their manuscript fully available?

Reviewer #1: Yes

Reviewer #2: Yes

5. Is the manuscript presented in an intelligible fashion and written in standard English?

Reviewer #1: (No Response)

Reviewer #2: Yes

6. Review Comments to the Author

Reviewer #1: I appreciate the authors' efforts in addressing several of the concerns raised in my initial review. The revised manuscript shows notable improvements in technical detail and methodological rigor. However, while some progress has been made, several critical issues remain insufficiently addressed and require further attention.

Improvements Acknowledged:

Enhanced Technical Implementation Details: The authors have significantly improved the technical documentation, particularly in describing the 5-step workflow, specific technology stack (Selenium, Beautiful Soup 4, NLTK, PythaiNLP), and BERT model training parameters with over 80% accuracy on 3,000+ datasets.

LLM Integration Clarification: The integration of Google Gemini 2.0 Flash with RAG methodology is now well-documented, including specific prompt examples and response formats that enhance reproducibility.

Statistical Validation: The addition of G*Power analysis (actual power = 0.81) provides appropriate justification for the sample size, addressing concerns about statistical adequacy.

Ethical Considerations: The inclusion of an ethics section demonstrates awareness of data privacy concerns and GDPR compliance.

Remaining Critical Issues:

System Architecture Depth: While Figure 1 has been updated, it still lacks the technical granularity expected for a platform paper. The diagram needs to show:

Detailed component interactions and data flow

Database schema and API integration points

Specific framework interconnections

Scalability and performance architecture

Evaluation Limitations Persist: Despite statistical justification, the evaluation methodology remains limited:

No comparative analysis with existing tools (even basic functionality comparison)

Missing system performance metrics (processing speed, throughput, scalability)

Lack of real-world deployment validation beyond controlled user studies

Research Contributions Need Clarification: The manuscript still struggles to articulate clear technical innovations beyond implementation. What specific algorithmic, architectural, or methodological advances does this work contribute to the field?

Limited Scope of Ethical Discussion: While appreciated, the ethics section remains superficial. Given the sensitive nature of health-related social media data, more detailed discussion of data anonymization techniques, consent mechanisms, and privacy-preserving methods is essential.

Specific Recommendations for Further Revision:

Technical Architecture: Provide a detailed system architecture diagram showing component interactions, data pipelines, and integration points. Consider including performance benchmarks and scalability analysis.

Comparative Evaluation: Even if commercial tools cannot be directly compared, consider comparing against baseline approaches or open-source alternatives in terms of accuracy, usability, or functionality.

Performance Analysis: Include system performance metrics such as processing time for different data volumes, memory usage, and scalability limits.

Research Positioning: More clearly articulate the novel contributions beyond making existing technologies accessible. What unique insights, methods, or technical solutions does this work offer?

Minor Issues Still Present:

Figure 4 dashboard screenshots remain difficult to read and should be higher resolution

The related work section could better position this work within the broader health informatics literature

Some methodological details about web scraping challenges and solutions could be expanded

Overall Assessment:

The authors have demonstrated good faith efforts to address reviewer concerns, particularly in technical documentation and statistical rigor. However, the fundamental issues of system architecture depth, evaluation comprehensiveness, and research contribution clarity require more substantial revision. The work has merit and addresses an important need, but needs additional development to meet publication standards for a technical systems paper.

Reviewer #2: (No Response)

7. PLOS authors have the option to publish the peer review history of their article (what does this mean?). If published, this will include your full peer review and any attached files.

Reviewer #1: No

Reviewer #2: No

---

## [Decision Letter · Decision Letter 2]

4 Aug 2025

PONE-D-25-21639R2Democratizing Social Media for Health Research: LLM-Powered Data Analytics Platform for NCDsPLOS ONE

Dear Dr. Intharah,

Thank you for submitting your manuscript to PLOS ONE. After careful consideration, we feel that it has merit but does not fully meet PLOS ONE’s publication criteria as it currently stands. Therefore, we invite you to submit a revised version of the manuscript that addresses the points raised during the review process.

We look forward to receiving your revised manuscript.

Kind regards,

Issa Atoum

Academic Editor

PLOS ONE

Journal Requirements:

**Additional Editor Comments:**

We appreciate your efforts in addressing the reviewers’ comments. However, Reviewer 1 has raised a critical concern regarding inconsistencies in your reported results, particularly between the abstract, conclusion, and Table 4. These discrepancies must be resolved before we can proceed further. To validate the summary claims and ensure consistency, you must provide the raw data specifically supporting the comprehension score comparisons between lay users and data scientists.

Reviewers' comments:

Reviewer's Responses to Questions

**Comments to the Author**

1. If the authors have adequately addressed your comments raised in a previous round of review and you feel that this manuscript is now acceptable for publication, you may indicate that here to bypass the “Comments to the Author” section, enter your conflict of interest statement in the “Confidential to Editor” section, and submit your "Accept" recommendation.

Reviewer #1: All comments have been addressed

2. Is the manuscript technically sound, and do the data support the conclusions?

Reviewer #1: No

3. Has the statistical analysis been performed appropriately and rigorously? 

Reviewer #1: Yes

4. Have the authors made all data underlying the findings in their manuscript fully available?

Reviewer #1: Yes

5. Is the manuscript presented in an intelligible fashion and written in standard English?

Reviewer #1: No

6. Review Comments to the Author

Reviewer #1: The original abstract (v1, 23 Apr 2025, L28 – 31) states that lay users achieved significantly higher dashboard-comprehension scores than data-science professionals; the revised abstract (v2, 24 Jul 2025, L45 – 49) claims the opposite, asserting superior comprehension by data scientists. These diametrically opposed statements cannot both be correct and therefore raise serious concerns about result consistency.

Moreover, the v2 abstract is internally inconsistent: Table 4 still shows higher mean comprehension for lay users (M = 4.1 ± 0.6) than for data scientists (M = 3.5 ± 0.7; t = 2.94, p = 0.004), contradicting the narrative presented in lines 45 – 49.

Please (i) explain these discrepancies and update either the abstract(s) or the Results section so that all statements align with the data, and (ii) provide the complete, original dataset together with the full preprocessing and analysis scripts (including software versions and random seeds) so the reported statistics can be independently verified.

7. PLOS authors have the option to publish the peer review history of their article (what does this mean?). If published, this will include your full peer review and any attached files.

Reviewer #1: No

---

## [Author Response · Author response to Decision Letter 3]

5 Aug 2025

The response to reviewers is attached.

---

## [Decision Letter · Decision Letter 3]

8 Aug 2025

PONE-D-25-21639R3Democratizing Social Media for Health Research: LLM-Powered Data Analytics Platform for NCDsPLOS ONE

Dear Dr. Intharah,

Thank you for submitting your manuscript to PLOS ONE. After careful consideration, we feel that it has merit but does not fully meet PLOS ONE’s publication criteria as it currently stands. Therefore, we invite you to submit a revised version of the manuscript that addresses the points raised during the review process.

We look forward to receiving your revised manuscript.

Kind regards,

Issa Atoum

Academic Editor

PLOS ONE

Journal Requirements:

**Additional Editor Comments:**

There are **still unresolved issues** including sample representativeness, comprehensive BERT performance metrics, objective evaluation of LLM-generated summaries, and improved figure clarity. I hope this revision fully addresses all remaining concerns to ensure acceptance.

Reviewers' comments:

Reviewer's Responses to Questions

**Comments to the Author**

1. If the authors have adequately addressed your comments raised in a previous round of review and you feel that this manuscript is now acceptable for publication, you may indicate that here to bypass the “Comments to the Author” section, enter your conflict of interest statement in the “Confidential to Editor” section, and submit your "Accept" recommendation.

Reviewer #1: All comments have been addressed

2. Is the manuscript technically sound, and do the data support the conclusions?

Reviewer #1: Yes

3. Has the statistical analysis been performed appropriately and rigorously? 

Reviewer #1: Yes

4. Have the authors made all data underlying the findings in their manuscript fully available?

Reviewer #1: Yes

5. Is the manuscript presented in an intelligible fashion and written in standard English?

Reviewer #1: Yes

6. Review Comments to the Author

Reviewer #1: Thank you for the effort you have invested in the first two revision rounds—methods and clarity have improved markedly. Before the manuscript can be accepted, please address the following remaining issues:

Sample representativeness – All 30 usability testers are statistics students (10 were also in the design interviews). Please discuss this limitation explicitly and outline plans to validate the tool with patients and frontline clinicians.

BERT performance reporting – Report precision, recall, F1 and a confusion matrix on a held-out test set, and describe your train/validation/test split (or cross-validation) strategy.

LLM-generated summary evaluation – Complement participant Likert ratings with an objective metric (e.g., ROUGE/BLEU or medical-entity coverage) or clinician expert review with inter-rater agreement.

Data & code availability – PLOS ONE requires open data at submission. Please deposit all anonymized raw data, analysis scripts and dashboard configuration files in a public repository and cite the DOI.

Figure 2 visual quality – The current visualization is too minimal; enhance labels, legend and contextual information so readers can interpret the results at a glance.

Addressing these points will complete the manuscript’s methodological transparency and usability. I look forward to reviewing your revised version.

7. PLOS authors have the option to publish the peer review history of their article (what does this mean?). If published, this will include your full peer review and any attached files.

Reviewer #1: No

---

## [Author Response · Author response to Decision Letter 4]

24 Aug 2025

All responses are in the Cover Letter

---

## [Decision Letter · Decision Letter 4]

1 Sep 2025

Democratizing Social Media for Health Research: LLM-Powered Data Analytics Platform for NCDs

PONE-D-25-21639R4

Dear Dr. Intharah,

We’re pleased to inform you that your manuscript has been judged scientifically suitable for publication and will be formally accepted for publication once it meets all outstanding technical requirements.

Kind regards,

Issa Atoum

Academic Editor

PLOS ONE

Additional Editor Comments (optional):

Reviewer #1:

Reviewers' comments:

Reviewer's Responses to Questions

**Comments to the Author**

1. If the authors have adequately addressed your comments raised in a previous round of review and you feel that this manuscript is now acceptable for publication, you may indicate that here to bypass the “Comments to the Author” section, enter your conflict of interest statement in the “Confidential to Editor” section, and submit your "Accept" recommendation.

Reviewer #1: All comments have been addressed

2. Is the manuscript technically sound, and do the data support the conclusions?

Reviewer #1: Yes

3. Has the statistical analysis been performed appropriately and rigorously? 

Reviewer #1: Yes

4. Have the authors made all data underlying the findings in their manuscript fully available?

Reviewer #1: Yes

5. Is the manuscript presented in an intelligible fashion and written in standard English?

Reviewer #1: Yes

6. Review Comments to the Author

Reviewer #1: I have carefully reviewed the authors' revised manuscript and their detailed response to the previous review comments. I am pleased to report that the authors have addressed all the concerns and suggestions raised in the initial review in a thorough and satisfactory manner.

7. PLOS authors have the option to publish the peer review history of their article (what does this mean?). If published, this will include your full peer review and any attached files.

Reviewer #1: No

---

## [Editor Report · Acceptance letter]

PONE-D-25-21639R4

PLOS ONE

Dear Dr. Intharah,

I'm pleased to inform you that your manuscript has been deemed suitable for publication in PLOS ONE. Congratulations! Your manuscript is now being handed over to our production team.

Kind regards,

on behalf of

Dr. Issa Atoum

Academic Editor

PLOS ONE